# Variability of Genetic Characters Associated with Probiotic Functions in *Lacticaseibacillus* Species

**DOI:** 10.3390/microorganisms10051023

**Published:** 2022-05-13

**Authors:** Franca Rossi, Carmela Amadoro, Maria Luigia Pallotta, Giampaolo Colavita

**Affiliations:** 1Diagnostica Specialistica, Sezione di Campobasso, Istituto Zooprofilattico Sperimentale dell’Abruzzo e del Molise “G. Caporale”, Via Garibaldi 155, 86100 Campobasso, Italy; 2Medicine and Health Science Department “V. Tiberio”, University of Molise, Via de Santis, 86100 Campobasso, Italy; carmela.amadoro@unimol.it (C.A.); pallottamarialuigia@gmail.com (M.L.P.); colavita@unimol.it (G.C.)

**Keywords:** *Lacticaseibacillus* species, probiotic potential, genetic traits, presence in genomes

## Abstract

This study aims to explore the intra-species distribution of genetic characteristics that favor the persistence in the gastrointestinal tract (GIT) and host interaction of bacteria belonging to species of the *Lacticaseibacillus* genus. These bacterial species comprise commercial probiotics with the widest use among consumers and strains naturally occurring in GIT and in fermented food. Since little is known about the distribution of genetic traits for adhesion capacity, polysaccharide production, biofilm formation, and utilization of substrates critically important for survival in GIT, which influence probiotic characteristics, a list of genetic determinants possibly involved in such functions was created by a search for specific genes involved in the above aspects in the genome of the extensively characterized probiotic *L. rhamnosus* GG. Eighty-two gene loci were retrieved and their presence and variability in other *Lacticaseibacillus* spp. genomes were assessed by alignment with the publicly available fully annotated genome sequences of *L. casei*, *L. paracasei*, *L. rhamnosus*, and *L. zeae*. Forty-nine of these genes were found to be absent in some strains or species. The remaining genes were conserved and covered almost all the functions considered, indicating that all strains of the genus may exert some probiotic effects. Among the variable loci, a taurine utilization operon and a α-L-fucosidase were examined for the presence/absence in 26 strains isolated from infant feces by PCR-based tests. Results were variable among the isolates, though their common origin indicated the capacity to survive in the intestinal niche. This study indicated that the capacity to exert probiotic actions of *Lacticaseibacillus* spp. depends on a conserved set of genes but variable genetic factors, whose role is only in part elucidated, are more numerous and can explain the enhanced probiotic characteristics for some strains. The selection of the most promising probiotic candidates to be used in food is feasible by analyzing the presence/absence of a set of variable traits.

## 1. Introduction

Bacteria of the *Lacticaseibacillus* genus, in particular the species *Lacticaseibacillus casei*, *paracasei*, *rhamnosus*, and *zeae*, comprise strains able to colonize the human gastrointestinal tract (GIT) and exert probiotic effects. These species are also adapted to fermented food environments, primarily dairy products, but also fermented foods of plant origin. Many studies have dealt with the demonstration of the beneficial effects of these bacteria and some of them exert probiotic effects able to ameliorate or prevent different medical conditions [1,2,3]. Among these, *L. rhamnosus* GG (ATCC 53103) is the most extensively studied [4]. These bacteria are commercially available as food supplements or in probiotic food products, but they also constitute the spontaneous microbiota of fermented foods of large consumption, including traditional products. In particular, *L. paracasei* strains constitute one of the main microbial components in traditional cheeses during and at the end of ripening. These foods can be a source of *Lacticaseibacillus* strains in numbers sufficient to influence host health [5]. However, strains that are part of the natural dairy microbiota could have a variable capacity to exert beneficial effects, so that it could be useful to select those with enhanced probiotic potential. Indeed, the selection, among the autochthonous microorganisms, of *Lacticaseibacillus* bacteria with traits associated to probiotic functions and use of these as added cultures, could confer health-promoting properties to the dairy products and also prevent the development of adventitious bacteria with undesirable characteristics.

Therefore, this study was focused on identifying genetic traits that, if present, can increase the potential of a bacterial strain belonging to the *Lacticaseibacillus* genus to behave as a probiotic. To this aim, the genome of the most-studied *Lacticaseibacillus* probiotic, *L. rhamnosus* GG, was used as a reference for identifying gene loci encoding for characteristics involved in survival in GIT and colonization capacity, adhesion to mucus or other host molecules, and production of cell-surface-associated macromolecules, including exopolysaccharides (EPS) involved in adhesion and immune modulation [6]. All the gene loci to which any of the above functions could be assigned, based on the existing annotation or on protein databases and scientific literature consultation, were searched by Blastn in the other *Lacticaseibacillus* spp. completely annotated genomes. The variability in the presence/absence of each gene, or gene cluster where appropriate, is presented. In addition, the presence/absence analysis of variable loci encoding taurine uptake and fucose utilization, as well as physiological features such as tolerance to bile salts and biofilm formation, were determined for 26 *Lacticaseibacillus* fecal isolates. These properties were selected on the basis of previous evidence on their involvement in survival in the intestinal environment. Indeed, taurine utilization capacity conferred by the *tau*BAC gene cluster was suggested to increase bile tolerance and persistence in GIT [7], while utilization of L-fucose, one of the most common monosaccharides in glycans on mammalian cell surfaces and intestinal mucus, allowed by the presence of α-L-fucosidases, is advantageous for use of this sugar as a carbon source in GIT [8].

## 2. Materials and Methods

### 2.1. Analysis of Lacticaseibacillus spp. Genomes for Genes Involved in Survival in GIT and Adhesion

The whole genome sequence of *L. rhamnosus* GG (GenBank acc. n. FM179322.1) was examined visually for the presence of genes involved in survival in GIT, polysaccharide production, and adhesion on the basis of the predicted function indicated for each gene locus. Membrane- and cell-surface-associated proteins without an assigned function were analyzed by Blastp (https://blast.ncbi.nlm.nih.gov/, accessed on 2 May 2022), Interpro (https://ebi.ac.uk/interpro/, accessed on 29 March 2022), and UniProt (https://www.uniprot.org, accessed on 29 March 2022) searches to derive a putative functional role. Only genes encoding proteins with a function assigned on the basis of the above analyses, or experimentally proven on the basis of scientific literature, were retained for the search of homologs of the encoding genes in the available complete and fully annotated genomes of *L. casei*, *L. paracasei*, *L. rhamnosus*, and *L. zeae* by Blastn. The latter species was considered for its close relatedness with *L. casei* and for its probiotic potential [9]. The cut-off values fixed for the definition of homology were a minimum of 30% query coverage and sequence identity above 60%.

Polysaccharide production gene clusters were graphically represented by using the https://katlabs.cc/genegraphics/app, accessed on 1 March 2022.

### 2.2. Bacterial Strains and Culture Conditions

Bacterial strains examined were isolated from the feces of children in a previous study [10] and assigned to the species *L. casei*, *L. paracasei*, *L. rhamnosus*, and *L. zeae* according to the highest Blastn scores of their 16S rRNA gene sequences. *L. rhamnosus* GG ATCC 53103 was used as positive control in PCR-amplification tests for genes *tau*B and the α-L-fucosidase gene LGG_02652. Lactobacilli were subcultured in an MRS broth or agar (Biolife Italiana, Milan, Italy) at 37 °C in aerobiosis for 48 h.

### 2.3. Molecular Techniques

DNA was extracted from 1 mL of fresh bacterial culture according to Amadoro et al. (2018). PCR tests for the screening of relevant genetic traits were carried out with primers 27f (5′-AGAGTTTGATCCTGGCTCAG-3′)/1492r (5′- GGTTACCTTGTTACGACTT-3′) targeted on the 16S rRNA gene, TauBF (5′-AGG(C/G)TC(G/T)GCATAGGC-3′)/TauBR (5′-CATGT(A/G)G(A/C)(C/T)TA(C/T)TGTTAC-3′) targeted on the taurine uptake gene *tau*B, locus LGG_00172, and FucF (5′-(G/T)AAC(C/G)ACCCAGTCACT-3′)/FucR (5′-G(A/T)CAGAACCA(C/T)TACCG) targeted on the α-fucosidase gene, locus LGG_02652. The latter two degenerate primer pairs were designed on consensus nucleotide positions in the genes *tau*B and LGG_02652 after alignment of the homologous gene sequences by Clustal Ω (https://www.ebi.ac.uk/Tools/msa/clustalo/, accessed on 10 September 2021) for *L. paracasei* and *L. rhamnosus* and all the *Lacticaseibacillus* species, respectively. Primer specificity was checked by Blastn and their melting temperature and tendency to form dimers were optimized by the Eurofins Genomics (Ebersberg, Germany) Oligo Calculator (https://www.google.com/search?client=firefox-b-d&q=eurofins+oligo+calculator, accessed on 12 September 2021).

The expected length of the amplification product was 649 for *L. paracasei* and 665 bp for *L. rhamnosus* for TauBF/TauBR primers and 1220 bp for FucF/FucR primers for both species. In the PCR reactions, primers were used in 0.5 µM final concentration carried out with the Takara Bio EmeraldAmp GT PCR Master Mix (Diatech, Jesi, AN, Italy). PCR programs comprised an initial denaturation at 94 °C for 5 min, 40 cycles of denaturation at 94 °C for 30 s, annealing for 30 s, and elongation at 72 °C for 1 min. A final elongation at 72 °C for 5 min was executed. The annealing temperatures were 55 °C for primer pair 27f/1492r and 50 °C for the other primer pairs.

The amplification products were separated on a 1.5% (*w*/*v*) agarose gel prepared in 1× TAE buffer (80 mM Tris/acetate, 2 mM EDTA, pH 8.0), stained with GelRed (Biotium, Fremont, CA, USA, DiaTech, Jesi, Italy) in the recommended amount, and run at 120 V in 1× TAE buffer.

Sequencing of the amplification products was carried out after purification with the Wizard^®^ SV Gel and PCR Clean-Up System (Promega Italia Srl, Milan, Italy) at Eurofins Genomics and the same primers used for amplification were used as sequencing primers.

### 2.4. Biofilm Forming Capacity

Two hundred µL of a 48 h culture in an MRS broth of each strain were transferred in triplicate in a microtiter plate well and incubated at 37 °C for 24 h. After the incubation the well contents were aspirated and the well was washed thrice with sterile saline. The well was filled with 200 µL of 99% methanol and kept for 5 min at room temperature. Methanol was aspirated and the well was let dry before adding 200 ul of a 2% (*w*/*v*) aqueous solution of crystal violet (Merck Life Science S.r.l., Milan, Italy). This solution was left in contact for 5 min. The colorant solution was removed; the wells were let dry and washed several times with water. Finally, 160 µL of 33% (*w*/*v*) acetic acid solution were added and the optical density (OD) of the wells was read at 620 nm in a 1420 multilabel counter Victor 3 v plate reader (PerkinElmer Italia, Milan, Italy).

### 2.5. Bile Salt Tolerance Test

Bile tolerance was assayed as described by Bustos et al., 2006 [11] by determination of the transmembrane electrical potential (Δψ) dissipation of energized cells, after the addition of bile salts to a 1.5% (*w*/*v*) final concentration. The method of Bustos et al., 2006 [11] was modified for use of safranin O (λ excitation 520 nm, λ emission 570 nm) 1.25 µM as a cell uptake probe and the use of 1 µM of protonophore carbonyl cyanide-p-trifluoromethoxy-phenyl hydrazine (FCCP) to completely depolarize cell membranes. The experiments were carried out in triplicate. Changes in fluorescence were measured with a LS50B spectrofluorimeter (PerkinElmer). The Δψ dissipation level caused by bile salts was expressed as a percentage of the increase in fluorescence after bile salt addition on the increase in fluorescence measured after total (100%) Δψ dissipation with FCCP.

### 2.6. Statistical Analyses

The data series of bile salt tolerance and biofilm formation were compared among strains by the Student’s *t*-test in Microsoft Excel. These were considered to be statistically distinct for *p* ≤ 0.05.

## 3. Results

### 3.1. Distribution of Genetic Traits Required for Probiotic Activity in Lacticaseibacillus Species

The selection of genetic traits to be compared among strains in the genome of *L. rhamnosus* GG, after identification of some proteins with unassigned function by search in protein databases, resulted in the list reported in Table 1.

It can be stated that at least one representative of each functional group of genes was found to be present in all genomes, except the bile salt hydrolase homologous to LGG_00501, genes for taurine utilization, sortases and pili, the anti-inflammatory protein PrtR encoded by locus LGG_02734, and fibrinogen-binding functions.

The role of each gene and the percentage of the analyzed genomes in which it is found is detailed in Appendix A. Each gene locus in the list was aligned by Blastn to the completely annotated genomes of the species *Lacticaseibacillus casei*, *L. paracasei*, *L. rhamnosus*, and *L. zeae*, which were represented by 5, 54, 38, and 2 entries, respectively, in NCBI at the time of analysis. Strain 12A was included in the analyses as a member of the *L. paracasei* species because it was found to be erroneously identified as *L. casei* [12].

Traits that are present in all genomes include the bile salt hydrolase (*bsh*) gene. This indicates the common ability to detoxify bile salts conferred by this enzyme in *Lacticaseibacillus* species [13]. An exception is *L. paracasei* ATCC 334, in which a truncated copy of the gene is present. Surface antigens p40, p75, p60, MetQ, and LGG_00503 are also conserved. However, the MetQ-encoding gene appears to be truncated in *L. paracasei* N1115, and therefore it is possibly not functional in this strain. Among these surface antigens, p40 and p75 have a role in the protection of inflammation and integrity of the intestinal epithelium [14] and in *L. paracasei* BL23 have cell-wall hydrolase activity and are secreted in microvesicles [15,16]. Antigen p60 has an immunomodulatory function [17] and the myosin-cross-reactive antigen encoded by locus LGG_00503 may be also involved in the production of conjugated linoleic acid [18]. Other conserved genes are 11 glycosyltransferases for the biosynthesis of polysaccharides or lipopolysaccharides, a flippase-like protein LGG_00827, a lipotheicoic acid (LTA) synthase LGG_00830, a polysaccharide biosynthesis transport protein LGG_00851, the LiaX daptomycin-sensing surface protein LGG_00914, a PspC domain-containing protein that in *Staphylococcus mutans* mediates biofilm formation in vivo [19], a toxin immunity protein LGG_01002, a lipopolysaccharide assembly protein LGG_01366, a fibronectin binding protein FbpA, a AI-2E family transporter possibly involved in biofilm formation, a mucus-binding protein MucBP, the first proteins of two gene clusters for polysaccharide production LGG_01990 and LGG_02036, a teichoic acid glycosylation protein LGG_02144, two PsaA putative adhesion lipoproteins and a polysaccharide transport protein LGG_02520.

The variable genes follow a species-specific or a strain-specific distribution. Namely, some strains of *L. casei* do not have a FeoB for Fe(II) uptake-encoding gene that confers increased gut colonization ability [20]. The *L. paracasei* species lacks a β-N-acetylhexosaminidase that is variable in *L. rhamnosus* strains, the TauE protein involved in taurine metabolism, the lectin-like protein LGG_00579, the cell surface protein LGG_00584, the MabA extracellular matrix binding protein, a modulator of adhesion and biofilm formation [21], the cell envelope-associated proteinase LGG_02734, lactocepin PrtR, able to selectively degrade pro-inflammatory chemokines and reduce inflammation in experimental IBD models [22], and the fibrinogen-binding protein LGG_02282. Some *L. paracasei* strains lack the InlJ internalin LGG_02337. *L. casei* and *L. zeae* lack genes encoding the SpaCBA and SpaFED pili, a pilin subunit LGG_00422, the adhesion exoprotein LGG_02923, and five proteins containing the WxL domain [23]. These proteins are involved in single-species biofilm formation and for some a lectin function was proven [24]. Other proteins not encoded in the genomes of *L. casei* and *L. zeae* are the extracellular complex proteins SpcA and B, the adhesin LGG_01590, one glycosyltransferase, and the cell surface protein LGG_00578. The cell-surface-docked proteins encoded by the gene cluster LGG_01589 to LGG_01592 that might be involved in adhesion, are highly conserved among S-layer-forming lactobacilli, are expressed constitutively, and are specific to vertebrate-adapted species, suggesting a role in adaptation to these hosts [25]. Worthy of note is that the SpaCBA pilus has a variable presence in *L. rhamnosus*, while it was found to be present in all analyzed *L. paracasei* strains, with six of them having an additional plasmid-encoded copy. In *L. paracasei*, an additional pilin subunit D1, SpaA (LGG_00422), present in all genomes, is also found on plasmid in six strains. The remaining genes with an intra-species varying distribution are the SpaFED pili, the taurine metabolism system, and proteins associated with EPS production. The role of the SpaFED pilus is not well-defined and it was reported not to be expressed in *L. rhamnosus* GG [26,27]. The EPS production genes in *L. rhamnosus* GG are arranged in three gene clusters, namely loci from LGG_00278 to LGG_00283, loci from LGG_01990 to LGG_02005 and loci from LGG_02036 to LGG_02054. The first cluster comprises genes present only in some *L. paracasei* and *L. rhamnosus* strains. These are three ramnosyltransferases, a polysaccharide transporter Eps1, and an Eps2 protein involved in polysaccharide biosynthesis that in some strains are dislocated in different EPS gene clusters. Given the high variability in EPS production gene arrangement, the remaining two clusters are shown graphically in Figure 1 for strains representative of the different gene content and order observed.

### 3.2. Testing of Genetic and Physiological Features in Lacticaseibacillus Strains from Feces

As relevant features for adaptation to the intestinal environment, taurine- and fucose-utilization capacity were tested in 26 strains identified as *Lacticaseibacillus* species by 16S rRNA gene sequencing among lactobacilli from the feces of children obtained in a previous study [10]. Moreover, bile salt resistance was determined by a fluorimetric method and biofilm formation capacity in microtiter plates was also examined. Results are summarized in Table 2.

It is possible to observe that taurine utilization and fucosidase activity presence are neither shared by all the intestinal isolates nor present in most of them. The amplification products for the PCR assays targeted on the *tau*B gene and on the α-fucosidase LGG_02652, whose identity was confirmed by the sequencing of the amplicons obtained from *L. rhamnosus* GG, are shown in Figure 2.

The strains formed four statistically distinct groups (a–d) on the basis of tolerance to bile salts and three strains, *L. paracasei* 6-15-1, *L. rhamnosus* Z-15-4, and *L. zeae* J-7-2, were not included in any group. Strain *L. zeae* J-7-2 was the most sensitive to bile salts based on the 95% extent of membrane depolarization.

According to biofilm-forming capacity, the strains were distributed in six statistically distinct groups (a–f) with strains *L. rhamnosus* G-7-14 and SA-7-6 not included in any of those groups. Four isolates, *L. casei* C-15-1b, *L. paracasei* P-7-13, *L. rhamnosus* AN-0-1, and G-7-16, showed a biofilm-forming capacity comparable to that of *L. rhamnosus* GG, while strains *L. paracasei* AN-15-1 and J-7-3 and *L. rhamnosus* G-7-14 and SA-7-6 exhibited a biofilm-forming capacity stronger than *L. rhamnosus* GG.

Examples of traces showing the increase in fluorescence intensity caused by the release of safranin O after the addition of bile salts to energized cells and complete depolarization after addition of FCCP are shown in Figure 3.

## 4. Discussion

This study highlighted that the genetic characteristics that influence survival and persistence in GIT and the probiotic effects exerted by bacteria of the *Lacticaseibacillus* genus are very complex. Genetic loci common to all the analyzed genomes encode for functions such as adhesion, bile resistance, polysaccharide production, fibronectin binding, and cell–cell signaling and represent a constant endowment of genes that allow coping with the GIT environment by all bacteria of the species considered. An exception among the analyzed genomes is represented by the dairy strain *L. paracasei* ATCC 334 (Acc. N. NC_008526.1), in which a nonfunctional *bsh* gene is present and one of the main EPS-production gene clusters is almost completely deleted (Figure 1B). Among the conserved genes the one encoding of the MetQ protein was also found to be truncated in one out of 54 *L. paracasei* genomes analyzed, thus indicating that in a minority of strains some of the most conserved traits may also be lacking.

Among the common traits, the *bsh* gene product has particular relevance for the probiotic function, since it has been shown to efficiently lower total and low-density lipoprotein cholesterol [28].

Another common trait that favors host colonization is the presence of the fibronectin-binding protein (FnBP) FbpA. This protein type binds fibronectin, a multidomain glycoprotein found in the human body fluids, extracellular matrices, and intestinal epithelial cells that are common targets for bacterial adhesins in GIT. The FbpA from *L. paracasei* BL23 has been characterized and found to exhibit a strong affinity for immobilized fibronectin [29]. In *L. acidophilus*, a mutant with inactivated *fbp*A, exhibited a significant decrease in adhesion to epithelial cells in vitro. While in pathogens, some FnBPs contribute to virulence, FnBPs in commensal and probiotic strains these proteins are essential for persistence in their ecological niches and might exert competition against pathogens for binding to fibronectin [30].

In most of the genomes, except for two *L. casei* strains, a FeoB protein, essential for the uptake of ferrous iron and gut colonization, is encoded [20].

Proteins p75 and p40 present in all *Lacticaseibacillus* genomes were found to mitigate intestinal inflammation through activation of the epidermal growth factor receptor [31] and upregulation of a proliferation-inducing ligand in the epithelium that stimulates the secretion of immunoglobulin A and relieves cytokine-induced apoptosis in the intestinal epithelial cells [32]. In addition, p75 and p40 stimulate epithelial cells to activate pathways that enhance their survival and barrier function to prevent bacterial translocation and the invasion by toxins [33].

The majority of the genetic determinants with a role in adaptation in GIT considered in this study was variable in the *Lacticaseibacillus* genomes. Among these, the sortase-dependent pili SpaCBA and SpaFEG, with the first having a proven role in adhesion to mucus, collagen, biofilm formation, and immune cell response stimulation, are included [26]. Similarly to what observed in this study, Douillard et al. [7] found that all the *L. paracasei* strains that they examined contained a SpaCBA pilus cluster. However, only the *L. rhamnosus* strains produced a functional SpaCBA pilus since the insertion of an IS 30 element upstream of the pilus gene cluster had constituted a strong promoter that allowed pilus expression. On the other hand, they observed that some strains displayed a mucus-binding capacity also in absence of SpaCBA pili, suggesting the existence of alternative mucus-binding mechanisms.

The relevance of the SpaCBA pilus in *L. paracasei* physiology should be better elucidated. This is present in two copies in some *L. paracasei* strains, among which strain LP10266 (Acc. n. NZ_CP031785.1 and n. NZ_CP031786.1) was recently shown to exhibit increased adhesion capacity. This strain was isolated from a patient with endocarditis and it was hypothesized that increased adherence can represent a virulence trait [34]. High adherence determined by a plasmid-encoded SpaCBA pilus was reported also for a *L. paracasei* strain isolated from raw cow’s milk [35].

The SpaCBA pilus was absent in the *L. casei* and *L. zeae* genomes analyzed, but this can be attributed to their low number, that is five genomes and two genomes, respectively. This might explain why the *tau*B gene, absent in the five *L. casei* genomes analyzed, was instead detected in the fecal isolates.

However, in general, the strains of these two species analyzed in this study are defective of many important adaptive traits. This might explain their infrequent occurrence in different ecological niches compared to *L. paracasei* and *L. rhamnosus*.

The taurine uptake system is another variable trait that, based on the genome analysis of 19 *L*. *casei/paracasei* strains, was proposed to be lost by strains adapted to dairy niches [36]. However, in this study, it appeared to be absent also in strains of intestinal origin, indicating that it may not be essential, at least in the short term, for survival in GIT. It can be noted that this gene, predicted to be absent in the *L. casei* genomes, was instead found to be present in the strains screened by PCR, thus indicating that the few genomes available for this species do not allow an exhaustive analysis of the distribution of genetic determinants relevant for probiotic action.

The α-L-fucosidase found to be absent in some genomes in this study and corresponding to the experimentally characterized LCABL_28270 in *L. paracasei* BL23 (GenBank acc. n. FM177140) (Rodríguez-Díaz et al., 2011 [8]) was also proposed to be lost by dairy strains [7]. However, it appeared to be present only in a minority of fecal isolates.

A great variability was displayed by the EPS-production gene clusters. This observation explains the diverse EPS types produced by *Lacticaseibacillus* strains and experimentally characterized [37,38]. Given the multiple actions exerted by these macromolecules, such as immunomodulation, antioxidant properties, enhancement of the hydrophobicity of bacterial cell surface that increases binding to the intestinal mucosa [37,38], variability in composition of these macromolecules can create also a variability of the effects that these bacteria can exert, that deserve to be defined to the strain level.

MucBP proteins were found to be present in most genomes. One of these, namely the LGG_02337, an adhesin distributed on the whole cell surface, participates in the adhesion of *L. rhamnosus* GG to mucus, and was recognized to be involved in pilus-mediated mucosal adhesion [25].

Finally, proteins with a WxL C-terminal domain, shown to possibly form a cell-surface protein complex involved in the degradation of plant polysaccharides in other lactobacilli, have a lectin-like function [24]. One protein of this group was found to be responsible for the adherence of *L. rhamnosus* GR-1 to the vaginal epithelium [39].

## 5. Conclusions

This study highlighted that all the *Lacticaseibacillus* spp. genomes analyzed comprise a common set of genes that could favor probiotic functions to be exerted by most of the microorganisms belonging to this genus. However, another set of gene loci is variable and can confer increased colonization capacity and beneficial interaction with the host to some strains. The presence of genetic traits relevant for survival in GIT, namely taurine and fucose utilization, as well as bile salt tolerance and biofilm formation, were found to be variable in fecal isolates, showing that a complex of features, not single traits, play a role in adaptation to the intestinal niche. Numerous traits emerged whose functional role is still little explored so far. This study resulted in the identification of variable genetic traits to be analyzed in the preliminary selection of the natural *Lacticaseibacillus* strain to be used to increase the beneficial properties of fermented products and probiotic candidates to be further characterized by whole-genome sequencing, in accordance with the current European food safety guidelines [40].

## Figures and Tables

**Figure 1 microorganisms-10-01023-f001:**
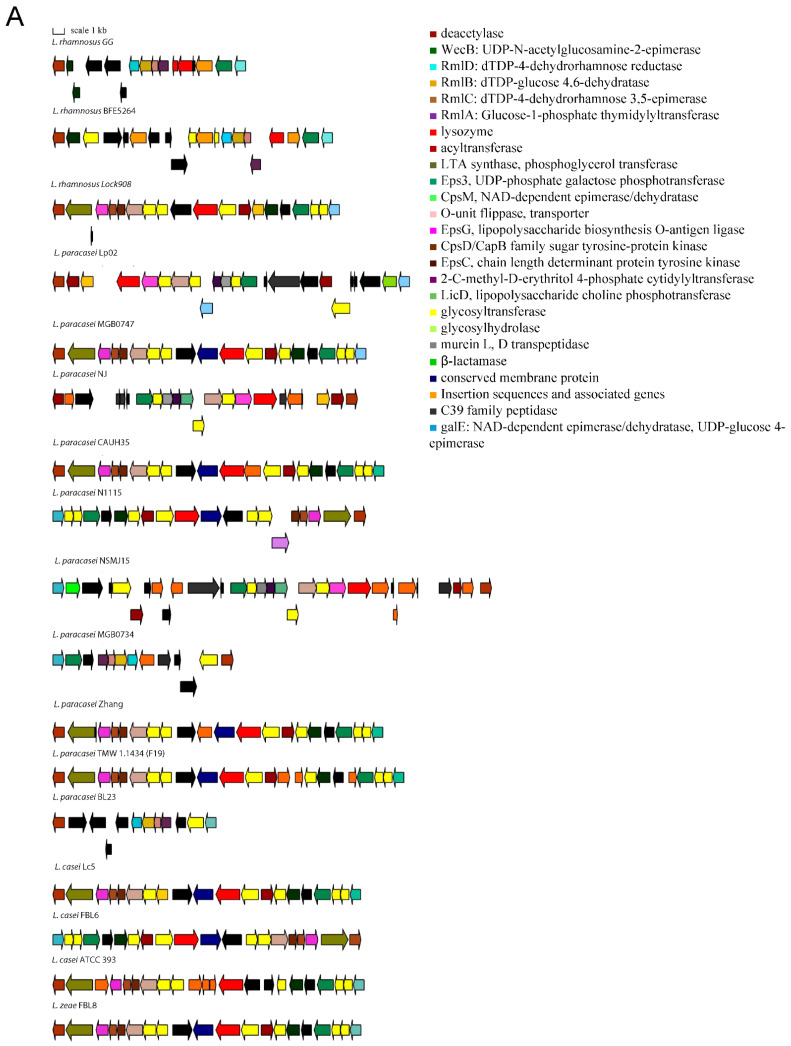
Gene arrangement in the EPS production gene clusters starting from proteins orthologous to LGG_01990 (**A**) and LGG_02036 (**B**).

**Figure 2 microorganisms-10-01023-f002:**
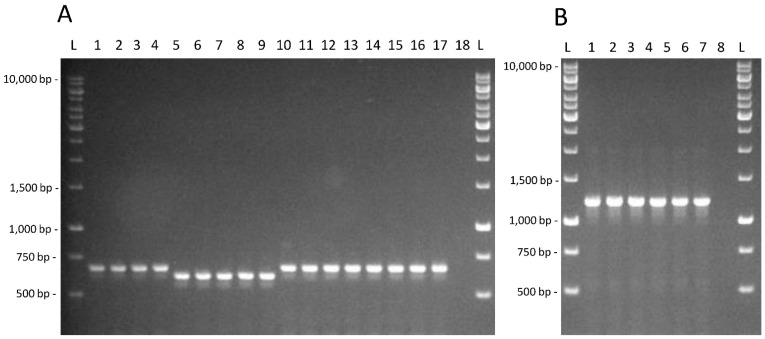
(**A**) Amplification products obtained for the *tau*B gene for strains *L. casei* AB-15-6 (1); C-15-1 (2); C-15-1b (3); G-0-6 (4); *L. paracasei* AN-15-2 (5); AN-15-3 (6); J-7-1 (7); J-15-4 (8); P-7-13 (9); *L. rhamnosus* AN-7-4 (10); AN-21-1 (11); D-0-5 (12); G-7-14 (13); G-7-16 (14); J-7-4 (15); Z-15-4 (16); GG (17); negative control (18). (**B**). Amplification products obtained for the α-fucosidase LGG_02652 targeted PCR test for strains *L. casei* C-15-1b (1); *L. paracasei* P-7-13 (2); *L. rhamnosus* AN-21-2 (3); D-0-5 (4); Z-15-4 (5); GG (6); negative control (7). L; GeneRuler 1 kb DNA Ladder (ThermoFisher Scientific, Monza, Italy).

**Figure 3 microorganisms-10-01023-f003:**
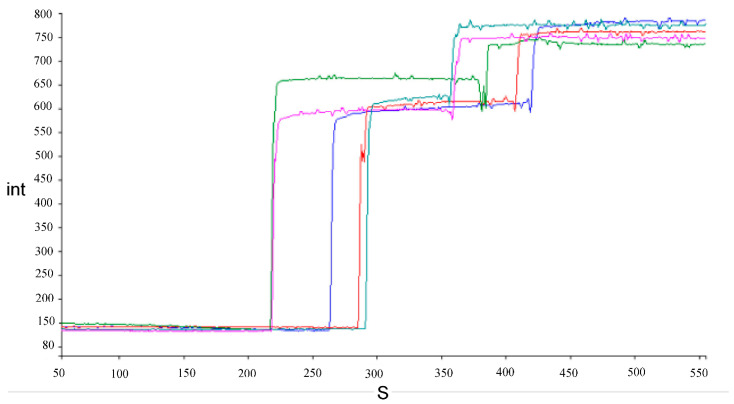
Examples of traces of fluorescence intensity (int) increase during time in seconds (S) for energized *Lacticaseibacillus* cells after addition of 1.5% (*w*/*v*) bile salts and FCCP in succession. Each color represents a different strain, namely, *L. paracasei* AN-15-3 (blue), J-7-1 (olive green) and J-15-4 (sea green), *L. rhamnosus* AN-21-2 (purple), and G-7-14 (red).

**Table 1 microorganisms-10-01023-t001:** Gene loci in the genome of *L. rhamnosus* GG encoding traits relevant for the capacity to exert probiotic properties separated on the basis of their functional role.

Function	Genetic Loci in *L. rhamnosus* GG
Anti-inflammatory protein	LGG_02734 ^b^
Bile salt hydrolase	LGG_00501 *
Biofilm formation	LGG_00914; LGG_01827
Cell wall anchored proteins with LPXTG domain	LGG_00434 ^b,f^; LGG_00578 ^c,d,e^; LGG_00584
Fibrinogen binding	LGG_01590 ^c,d,e,f^; LGG_02282 ^b^
Fibronectin binding	LGG_0005 ^a,b^; LGG_01450
Fucose utilization	LGG_02652 ^a^
Gut colonization	LGG_01877 ^g^
Lectin-like protein	LGG_00183; LGG_00576 ^a^; LGG_00579 ^b,c,d^; LGG_00583 ^c,d,e,f^; LGG_00585 ^c,d,e,f^; LGG_00587 ^c,d,e,f^; LGG_01765 ^c,d^
Mucus binding	LGG_01883; LGG_02337^e^
Pilus protein	LGG_00422 ^c,d,f^; LGG_00442 ^c,d,f^; LGG_00443 ^c,d,f^; LGG_00444 ^c,d,f^; LGG_02370 ^c,d^; LGG_02371 ^c,d^; LGG_02372 ^c,d^
Polysaccharide biosynthesis	LGG_00107 ^a,b,c,d^; LGG_00278 ^a,c,d^; LGG_00279 ^a,c,d^; LGG_00280 ^a,c,d^; LGG_00281 ^a,c,d^; LGG_00282 ^a,c,d^; LGG_00283 ^a,c,d^; LGG_00295; LGG_00348 ^c,d,e^; LGG_00349 ^e^; LGG_00645; LGG_00695 ^e^; LGG_00696; LGG_00697 ^e^; LGG_00825; LGG_00826; LGG_00827; LGG_00830; LGG_00851; LGG_00998; LGG_00999; LGG_01057; LGG_01066; LGG_01069; LGG_01147; LGG_01366; LGG_01538 ^a,d^; LGG_01586 ^e^; LGG_01587 ^c,d,e^; LGG_01990 †; LGG_02036 †; LGG_02144; LGG_02520; LGG_02869 ^h^
Sortase	LGG_00441 ^c,f^; LGG_02369 ^c,d^
Surface adhesin	LGG_01591 ^e,f^; LGG_01592 ^c,d,e,f^; LGG_01865 ^b^; LGG_02423; LGG_02426; LGG_02923 ^c,d,f^
Surface antigen	LGG_00031; LGG_00324; LGG_00500 *; LGG_00503; LGG_01589 ^a,c,d^; LGG_02016
Taurine utilization	LGG_00172 ^a,c,d^; LGG_00173 ^a,c,d^; LGG_00174 ^a,c,d^; LGG_00544 ^b^
Toxin immunity	LGG_01002

^a^ variable in all the *Lacticaseibacillus* species considered; ^b^ absent in *L. paracasei*; ^c^ absent in *L. casei*; ^d^ absent in *L. zeae*; ^e^ variable in *L. paracasei*; ^f^ variable in *L. rhamnosus*; ^g^ variable in *L. casei*; ^h^ variable in *L. zeae*; * truncated (pseudogene) in some strains; † conserved first gene in highly variable EPS production gene cluster.

**Table 2 microorganisms-10-01023-t002:** Distribution of variable genes involved in survival in GIT in intestinal isolates of *L. casei*, *L. paracasei*, *L. rhamnosus*, and *L. zeae*, percent dissipation of membrane potential in presence of 1.5% bile salts and biofilm formation extent.

Strain	Taurine Utilization	α-L-Fucosidase LGG_02652	% Membrane Potential Dissipation	Biofilm Formation (OD 620 nm)
*L. casei*				
AB-15-6	+		77 ± 0.5 ^a^	0.028 ± 0.01 ^a^
C-15-1	+		75 ± 0.5 ^a^	0.034 ± 0.01 ^a^
C-15-1b	+	+	83.5 ± 0.5 ^b^	0.160 ± 0.05 ^b^
G-0-6	+		86 ± 1 ^c^	0.065 ± 0.02 ^c^
*L. paracasei*				
AN-15-1			87 ± 0.5 ^c^	0.436 ± 0.05 ^d^
AN-15-2	+		77 ± 1 ^a^	0.122 ± 0.05 ^e^
AN-15-3	+		80 ± 0.5 ^d^	0.051 ± 0.02 ^f^
AN-15-4			87 ± 0.5 ^c^	0.068 ± 0.01 ^c^
J-7-1	+		83 ± 1 ^b^	0.115 ± 0.03 ^e^
J-7-3			79 ± 0.5 ^d^	0.534 ± 0.09 ^d^
J-15-4	+		75 ± 1 ^a^	0.034 ± 0.02 ^a^
P-7-13 ¤	+	+	77 ± 0.5 ^a^	0.133 ± 0.02 ^b^
6-15-1			71.5 ± 0.5	0.043 ± 0.01 ^f^
*L. rhamnosus*				
AN-0-1			83 ± 0.5 ^b^	0.171 ± 0.04 ^b^
AN-7-1			76 ± 1 ^a^	0.111 ± 0.05 ^e^
AN-7-4	+		87 ± 0.5 ^c^	0.119 ± 0.03 ^e^
AN-21-1	+		82.5 ± 0.5 ^b^	0.032 ± 0.01 ^a^
AN-21-2		+	81 ± 1 ^d^	0.076 ± 0.02 ^c^
C-0-4			75 ± 1 ^a^	0.025 ± 0.01 ^a^
D-0-5	+	+	77 ± 0.5 ^a^	0.048 ± 0.02 ^f^
G-7-14	+		80 ± 0.5 ^d^	0.270 ± 0.05
G-7-16	+		83 ± 0.5 ^b^	0.180 ± 0.02 ^b^
J-7-4	+		82 ± 0.5 ^b^	0.023 ± 0.01 ^a^
SA-7-6			83 ± 0.5 ^b^	0.613 ± 0.07
Z-15-4	+	+	73 ± 0.5	0.024 ± 0.01 ^a^
*L. zeae*				
J-7-2			95 ± 1	0.084 ± 0.02 ^c^
*L. rhamnosus* GG	+	+	79 ± 0.5 ^d^	0.138 ± 0.03 ^b^

Groups of data not statistically distinct for *p* ≤ 0.05 have the same apex; this is a, b, c or d for membrane potential dissipation and a, b, c, d or f for biofilm formation.

## Data Availability

Not applicable.

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
