# Peer review of "Variability of Genetic Characters Associated with Probiotic Functions in Lacticaseibacillus Species"

_microorganisms, 2022, doi:10.3390/microorganisms10051023_

Round 1

Reviewer 1 Report

Exploring the intra-species distribution of genes found in the genomes of bacteria from Lacticaseibacillus genus, the authors first compiled a list of genetic traits involved in adhesion, extracellular polysaccharide production, biofilm formation and utilization of substrates critical for survival in GIT in the genome the acknowledged probiotic L. rhamnosus GG. They then evaluated the presence of each gene in the genomes of other bacteria of Lacticaseibacillus genus and found that 32 of 81 gene loci are present in all isolates. Thus, the authors made an important conclusion that all strains of the genus may exert some probiotic effects, although their combinations are variable. Focusing on the EPS production genes, the authors demonstrated the variability of their genomic arrangements in two clusters, whereas the presence of taurine utilization operons and α-L-fucosidase genes was experimentally assessed in 26 strains isolated from infant feces. In the last part of the study, they evaluated biofilm-forming capacity and bile salt tolerance of the novel isolates. As a whole, the paper contains an important information, but requires essential improving of its presentation. In general, the article contains important information, but it requires a significant improvement in its presentation.

Major concerns

  1. Genes potentially associated with the probiotic activity of Lacticaseibacillus were selected to analyze their presence/absence in particular genomes. The information obtained has been summarized in Supplementary Table 1 and discussed in lines 157-210. (1) It is likely that the Table contains 81 rather than 82 loci as stated in Summary, so please check that out. (2) The data presented are important, but difficult to perceive. I think it would be better if in both cases the authors rearrange the presentation according to the target categories (adhesion, polysaccharide production, biofilm formation and utilization of substrates critical for survival). The table will immediately show whether the absence of some genes is compensated for by others with the same function or not, and it will be easier to follow the discussion in the main text.

  1. Lines 159-161: ]. “Surface antigens p40, p75, p60, MetQ and LGG_00503 are also conserved” but p75 and MetQ are much more truncated than the discussed bsh gene. How can you be sure that they keep their function?

  1. Figure 2 exemplifies the traces showing the loss of cell fluorescence intensity in the presence of bile salts, while the percentage of bile-mediated dissipation of the membrane potential is given numerically for all strains in Table 1. Please decipher “S” and “int” in Fig. 2 and specify in the legend isolates that were used as samples in this particular experiment.

  1. From 2.3 section in the Materials and Methods, it is not clear how amplicons were analyzed. It is likely that it was a simple fractionation on agarose gels that gave the expected band (indicated by “+” in Table 1) or not. However, it is also important to know how large the observed bands were compared to the reference strain. Therefore, I consider it essential to show the original gels for both genes in Figure 3, remove this information from Table 1 and discuss it accordingly.

  1. Lines 25-26: I am not sure that there is sufficient evidence for the main conclusion that “the capacity to exert probiotic actions of Lacticaseibacillus spp. depends on a conserved set of genes” (see explanation in point 9).

Minor concerns:

  1. Line 98: Please explain why you chose aerobic culture conditions that are not ideal for Lacticaseibacillus even in MRC media.
  2. Lines 103-106 Degenerate characters must be decoded
  3. Lines 153-155: “The strain labelled as casei 12A was excluded from the analyses since it gave, for all analyzed loci, percentages of nucleotide identity and query coverage values dissimilar from the other L. casei strains and similar to the values obtained for L. paracasei, so that it is possibly misidentified.” The belonging of strain 12A to L. paracasei has recently been proven (Frolova et al. 2021 Life, https://doi.org/10.3390/life11111246).
  4. Lines 256-257: The sentence “This study highlighted that all the Lacticaseibacillus genomes analyzed comprise a common set of genes that could favor probiotic functions to be exerted by most of the microorganissms belonging to this genus” is misleading. In fact, you only got 32 “common genes” by analyzing 81 genomes. If the number of analyzed genomes is increased (for example, by only adding 12A), the size of the “common set” may decrease.

Typos:

  1. Line 16 “… in the genome the extensively characterized probiotic rhamnosus GG”: missing “of” in front of “extensively”.

11. Lines 18-21: Consecutive statements “The presence/absence and variability of each gene…..”,   “…49 of these were found to be absent in some genomes….” and “A set of genes was found to be conserved…” are confusing. Specify, which gene set is conserved? 

Author Response

C) Major concerns

Genes potentially associated with the probiotic activity of Lacticaseibacillus were selected to analyze their presence/absence in particular genomes. The information obtained has been summarized in Supplementary Table 1 and discussed in lines 157-210. (1) It is likely that the Table contains 81 rather than 82 loci as stated in Summary, so please check that out. (2) The data presented are important, but difficult to perceive. I think it would be better if in both cases the authors rearrange the presentation according to the target categories (adhesion, polysaccharide production, biofilm formation and utilization of substrates critical for survival). The table will immediately show whether the absence of some genes is compensated for by others with the same function or not, and it will be easier to follow the discussion in the main text.

R) Thanks to the reviewer for the very useful criticisms and suggestions for revision.

We have checked the number of genes in Table S1 and the count 82 was confirmed.

To render more clear if the absence of some genes is compensated for by others with the same function or not we introduced a new table in the main text (Table 1 in the current version) in which functional groups of genetic loci are separated and apexes indicate the distribution of each locus in the different species, as explained in the footnotes. A statement regarding the distribution of functional traits among strains was also added in the text (Lines 180 – 183).

C) Lines 159-161: ]. “Surface antigens p40, p75, p60, MetQ and LGG_00503 are also conserved” but p75 and MetQ are much more truncated than the discussed bsh gene. How can you be sure that they keep their function?

R) thank you for this observation. We found that for MetQ but the same cannot be stated for p75. We added comments on MetQ (Lines 194 – 195; 306 - 310).

C) Figure 2 exemplifies the traces showing the loss of cell fluorescence intensity in the presence of bile salts, while the percentage of bile-mediated dissipation of the membrane potential is given numerically for all strains in Table 1. Please decipher “S” and “int” in Fig. 2 and specify in the legend isolates that were used as samples in this particular experiment.

R) the interpretation of the membrane potential dissipation experiment is now improved according to a newly added literature source (Bustos et al. 2011, ref. 11) reporting the first application of the method to lactobacilli and the explanation at lines 152 - 155. The meaning of the axes titles and the strains presented are now explained in the legend of Figure 2.

C) From 2.3 section in the Materials and Methods, it is not clear how amplicons were analyzed. It is likely that it was a simple fractionation on agarose gels that gave the expected band (indicated by “+” in Table 1) or not. However, it is also important to know how large the observed bands were compared to the reference strain. Therefore, I consider it essential to show the original gels for both genes in Figure 3, remove this information from Table 1 and discuss it accordingly.

B) Sorry for the omission; we added the method of separation on agarose gel (Lines 125 -127). Table 2 (previously Table 1) was not modified because it summarizes all the results for the strains that we analyzed. The bands amplified from the isolates positive for the genes are shown in the newly added Figure 2 (Lines 270 - 273).

C) Lines 25-26: I am not sure that there is sufficient evidence for the main conclusion that “the capacity to exert probiotic actions of Lacticaseibacillus spp. depends on a conserved set of genes” (see explanation in point 9).

R) we revised the comments according to this observation and on the basis of what emerged from Table 1 (Line 23, Lines 27 – 29; 180 – 183; 307 – 310).

C) Minor concerns:

R) Line 98: Please explain why you chose aerobic culture conditions that are not ideal for Lacticaseibacillus even in MRC media.

C) These bacteria are defined as oxigen tolerant anaerobics as from the publication with DOI: 10.1371/journal.pone.0099189. We agree that for some strains growth in anaerobiosis is more vigorous on plate, but in this case we chose aerobic conditions because we just had to subculture the strains. Colonies in plate could be equally isolated to prepares cultures in broth and extract DNA. Growth in broth was very similar in aerobiosis and anaerobiosis.

C) Lines 103-106 Degenerate characters must be decoded

R) done as requested (Lines 106 - 109)

C) Lines 153-155: “The strain labelled as casei 12A was excluded from the analyses since it gave, for all analyzed loci, percentages of nucleotide identity and query coverage values dissimilar from the other L. casei strains and similar to the values obtained for L. paracasei, so that it is possibly misidentified.” The belonging of strain 12A to L. paracasei has recently been proven (Frolova et al. 2021 Life, https://doi.org/10.3390/life11111246).

R) we commented accordingly (Lines 188 – 189) and added the reference suggested

C) Lines 256-257: The sentence “This study highlighted that all the Lacticaseibacillus genomes analyzed comprise a common set of genes that could favor probiotic functions to be exerted by most of the microorganissms belonging to this genus” is misleading. In fact, you only got 32 “common genes” by analyzing 81 genomes. If the number of analyzed genomes is increased (for example, by only adding 12A), the size of the “common set” may decrease.

R) adding that strain did not change the common set of genes and the percentages of presence of all genes in the species paracasei.

C) Typos:

Line 16 “… in the genome the extensively characterized probiotic rhamnosus GG”: missing “of” in front of “extensively”.

R) added (Line 18)

C) 11. Lines 18-21: Consecutive statements “The presence/absence and variability of each gene…..”, “…49 of these were found to be absent in some genomes….” and “A set of genes was found to be conserved…” are confusing. Specify, which gene set is conserved?

R) this part was rephrased (lines 19 - 23)

Reviewer 2 Report

Stability of genetic information is a prerequisite to use new  probiotics both in food products, as an ingredient or raw material, and in therapeutic approaches as cell factories of bioactive compounds. In this manuscript this study was focused on identifying genetic traits that, if present, can increase the potential of a bacterial strain belonging to the Lacticaseibacillus genus to behave as probiotics. Such research is very important for drug and food manufacturers. However, I have a few comments to the authors:

  1. Figure 2- what does “S” and “int” mean?
  2. Information on "% membrane potential dissipation" was not included in the methods
  3. At 570 nm was the OD determined at 620nm? They are given different in the methods and table.1
  4. Statistical analysis, eg Tukey's test, should be performed for the "% membrane potential dissipation" and for the OD value. Information on statistical analysis should also be included in the research methods.

Author Response

Reviewer 2

C) I have a few comments to the authors:

Figure 2- what does “S” and “int” mean?

R) We are thankful to the reviewer for helping to improve this manuscript.

The required information on figure axis titles was added in the legend to Figure 3 in the revised version.

C) Information on "% membrane potential dissipation" was not included in the methods

At 570 nm was the OD determined at 620nm? They are given different in the methods and table.1

R) sorry for the mistake, the correct wavelength is now reported also in the text (Line 142). A reference (Bustos et al., 2011) was added for the method and the % membrane potential dissipation measurement is explained at Lines 154 – 156.

C) Statistical analysis, eg Tukey's test, should be performed for the "% membrane potential dissipation" and for the OD value. Information on statistical analysis should also be included in the research methods.

R) Done as suggested. We compared data by using the Student's t test (Lines 157 – 160, Table 2).

Reviewer 3 Report

A very interesting and highly informative manuscript on the variability of the genetic characters associated with probiotic functions that is observed in Lacticaseibacillus species. The manuscript is very well written and only a few minor improvements can be suggested.

l. 46-47 and especially the part ‘so that…uncontrolled’. please rephrase to improve clarity

l. 105, 108, 110, 111, 115, 116, gene and species names should be written in italics

l. 107, it should read ‘degenerate’

l. 119-120, is there a final amplification step in the PCR program?

l. 170, please write the genus abbreviated as ‘S.’ in full, it is the first mention

figure 2, please explain what is shown in both axis and what the different colors indicate

l. 293, please consider replacing word ‘resulted’ with ‘was’

Author Response

Reviewer 3

C) The manuscript is very well written and only a few minor improvements can be suggested.

l. 46-47 and especially the part ‘so that…uncontrolled’. please rephrase to improve clarity

R) we are very grateful to the reviewer for appreciating our work. As recommended, the part was rephrased to improve clarity (Lines 48 - 49)

C) l. 105, 108, 110, 111, 115, 116, gene and species names should be written in italics

R) done accordingly throughout the manuscript

C) l. 107, it should read ‘degenerate’

R) this was corrected (Line 110)

C) l. 119-120, is there a final amplification step in the PCR program?

R) yes, sorry for the omission. This is now specified (Lines 122 - 123)

C) l. 170, please write the genus abbreviated as ‘S.’ in full, it is the first mention

R) done as required (Line 205)

C) figure 2, please explain what is shown in both axis and what the different colors indicate

R) the required explanations were added in the legend to Figure 3

C) l. 293, please consider replacing word ‘resulted’ with ‘was’

R) done as suggested (Line 334)

Round 2

Reviewer 1 Report

All my comments have been taken into account. Therefore, I think that the paper can be published, although some minor concerns appeared in the modified sections, and one ambiguity I missed in the previous version.

Minor concerns:

  1. Lines 282-287: It's a good idea to divide the strains into 7-8 different categories. Please indicate them in the text: “The strains were formed seven statistically distinct groups (a-g) on…” and “For 286 this characteristic the strains were distributed in eight statistically distinct groups (a-h)”.        
  2. Line 282: “The strains were formed seven statistically distinct groups…”. The word "statistically" should be omitted as some groups contain only 1 sample. The reliability of their difference from other groups cannot be established.
  3. Table 2: according to biofilm formation, the isolate AN-7-4 (OD = 0.119) should probably be assigned to the group “e”, rather than “d”.
  4. Lines 285-286: “…while the remaining isolates showed a biofilm forming capacity comparable to that of rhamnosus GG (n. 8) or lower (n. 15)”. What do “(n. 8) and (n. 15)” mean?
  5. Lines 397-398: Remove the last sentence.

Author Response

  1. C) All my comments have been taken into account. Therefore, I think that the paper can be published, although some minor concerns appeared in the modified sections, and one ambiguity I missed in the previous version.

Minor concerns:

  1. Lines 282-287: It's a good idea to divide the strains into 7-8 different categories. Please indicate them in the text: “The strains were formed seven statistically distinct groups (a-g) on…” and “For 286 this characteristic the strains were distributed in eight statistically distinct groups (a-h)”.        
  2. Line 282: “The strains were formed seven statistically distinct groups…”. The word "statistically" should be omitted as some groups contain only 1 sample. The reliability of their difference from other groups cannot be established.
  3. Table 2: according to biofilm formation, the isolate AN-7-4 (OD = 0.119) should probably be assigned to the group “e”, rather than “d”.
  4. Lines 285-286: “…while the remaining isolates showed a biofilm forming capacity comparable to that of rhamnosus GG (n. 8) or lower (n. 15)”. What do “(n. 8) and (n. 15)” mean?
  1. R) all the comments regarding the newly added statistical elaboration of bile tolerance and biofilm formation data were dealt according to the reviewer’s observations (Lines 282 – 284; 286 – 291). We are grateful to the review for carefully checking it.

Strain AN-7-4 was allotted to group e, as correctly suggested (it was an error in apex attribution) (Table 2)

C)

  1. Lines 397-398: Remove the last sentence.
  1. R) sorry for the oversight, the sentence was removed